# Accurate Identification of ADHD among Adults Using Real-Time Activity Data

**DOI:** 10.3390/brainsci12070831

**Published:** 2022-06-26

**Authors:** Amandeep Kaur, Karanjeet Singh Kahlon

**Affiliations:** 1Department of Computer Engineering and Technology, Guru Nanak Dev University, Amritsar 143005, Punjab, India; 2Department of Computer Science, Guru Nanak Dev University, Amritsar 143005, Punjab, India; karankahlon@gndu.ac.in

**Keywords:** ADHD, motor activity, diagnosis, actigraphic, PCA, SVM, classification, machine learning

## Abstract

Attention Deficit Hyperactivity Disorder (ADHD) is a neurodevelopment disorder that affects millions of children and typically persists into adulthood. It must be diagnosed efficiently and consistently to receive adequate treatment, otherwise, it can have a detrimental impact on the patient’s professional performance, mental health, and relationships. In this work, motor activity data of adults suffering from ADHD and clinical controls has been preprocessed to obtain 788 activity-related statistical features. Afterwards, principal component analysis has been carried out to obtain significant features for accurate classification. These features are then fed into six different machine learning algorithms for classification, which include C4.5, kNN, Random Forest, LogitBoost, SVM, and Naive Bayes. The detailed evaluation of the results through 10-fold cross-validation reveals that SVM outperforms other classifiers with an accuracy of 98.43%, F-measure of 98.42%, sensitivity of 98.33%, specificity of 98.56% and AUC of 0.983. Thus, a PCA-based SVM approach appears to be an effective choice for accurate identification of ADHD patients among other clinical controls using real-time analysis of activity data.

## 1. Introduction

ADHD (Attention Deficit Hyperactivity Disorder) is a prevalent childhood disorder characterized by inattention, hyperactivity, and impulsivity [1]. Children with ADHD have poor behavioral regulation and control, causing them to react inappropriately to a variety of stimuli. The incidence rate for school-age children globally is about 5%; approximately 80–85 percent of ADHD tweens tend to face difficulties into adolescence, and 60% encounter symptoms in adults [2]. ADHD can have a significant impact on academic attainment and social interactions, as it makes it difficult for the patients to focus their attention and manage their behavior [3,4]. Adult ADHD frequently has a more diverse clinical presentation that extends beyond the basic motor symptoms seen in children and includes a broader range of emotional dysregulation and functional impairment [5]. Adult ADHD tends to be characterized by inattention, which manifests itself as disorganization and distraction-, task- and focus-related difficulties, as well as a proclivity for daydreaming [6]. Adult ADHD is usually linked to other mental conditions such as depression, mood, anxiety and sleeplessness [7,8]. Moreover, the symptoms of ADHD frequently coincide with those of other psychiatric illnesses, such as affective and internalizing disorders, autism spectrum disorder, and autistic-like features [9]. There may be external factors that raise the chance of ADHD, such as exposure to chemicals and maternal lifestyle throughout pregnancy, but these characteristics are not prevalent in all human beings [10]. Furthermore, the disorder’s occurrence cannot be linked to a particular biomarker, such as a single faulty gene [11].

While males are more likely to be diagnosed with ADHD and the percentage of them receiving treatment is high, the existing research trends show that prevalence has been rising amongst females and people of various races over the last century [12]. Although females have lesser incidence rates than males, they have severe impairments associated with their diagnoses that necessitate early detection to minimize the chances of psychological illnesses, self-harm and suicidality [13]. The lack of clear ADHD diagnosis is due to a lack of understanding of the brain mechanisms that cause the disorder. The severity of ADHD symptoms, IQ, co-morbidities, family conflict or environmental discomfort, earlier care for ADHD, and the incidence of concurrent diseases are all significant factors that influence the trajectory of ADHD [6,14]. In the diagnosis of this disorder, stimulants are undoubtedly regarded as the most promising treatment, however, they carry a high risk of misuse, addiction, and pose different side effects such as loss of appetite, high blood pressure, heart problems, mood disorders, etc., [1]. However, there exist strong associations between adult ADHD with a higher risk of substance abuse [15]. The current way of diagnosing ADHD is a subjective evaluation and thorough clinical observation which are mostly imprecise and erroneous, so there is a need for more objective methods. Since the diagnosis is exclusively based on observed behavior and reported symptoms, there is a possibility of over-and-under-diagnosis and also there are no valid objective tests to identify ADHD [16].

Nowadays, sensory data gathered from patients and evaluated using machine learning approaches has attracted a lot of attention as a way to supplement existing subjective diagnostic practices in mental health, particularly ADHD [17]. Recent technological advancements, notably in the field of wearable devices, have enabled unparalleled access to physiological data. Moreover, wearable devices are employed in a variety of applications such as analyzing physical health and mental health. The studies using actigraph data from both adolescents and adults have found that motor activity has a significant role to play in the ADHD diagnosis. Moreover, HRV has been researched concerning ADHD, although the results are relatively less promising [1]. Generally, Electrocardiogram (ECG) [18], electroencephalogram (EEG) [19,20], magnetic resonance imaging (MRI) [21,22], game simulators [23], accelerometers [24] and many other methods are available for collecting sensory data. In practice, research is also carried out to analyze the EEG signal in the field of concentration of attention in mothers and children [25]. However, the activity data collected from actigraphic devices are considered as a gold standard for detecting different patterns such as sleep and awake states. The results derived from activity data are more precise, accurate, and reliable [26].

In light of these considerations, the present study presents a Support Vector Machine (SVM)-based approach for accurately identifying patients with ADHD among other clinical controls using activity data analysis. This approach makes use of statistical features related to the motor activity of patients, which are appropriately selected using Principal Component Analysis (PCA). The comparative analysis of the given SVM classifier with other state-of-the-art classifiers viz; Random Forest, C4.5 Decision tree, k-Nearest Neighbor, Naïve Bayes and LogitBoost, evidently reveals that the proposed PCA- and SVM-based activity features approach is an effective method for precise detection of ADHD patients among other patients suffering from different disorders. 

This paper has been organized into various sections: Section 2 introduces the key contributions in this field. The proposed framework and study approaches are elaborated in Section 3. Section 4 addresses the classification models to be employed along with their hyper-parameter tuning. The results drawn are presented in Section 5. While Section 6 discusses the research findings and outlines the current work’s constraints and potential future directions. Finally, the brief conclusions reached from the work’s primary outcomes are explained in Section 7.

## 2. Literature Survey

The goal of this systematic literature review is to evaluate the most relevant research on ADHD Diagnosis utilizing activity-related data and its analysis using machine learning techniques. A variety of search engines and databases were used, including Google Scholar, PubMed, IEEE Explore, Science Direct and others. The keywords were input as a Boolean search string that included (‘ADHD’ or ‘attention deficit hyperactivity disorder’), (‘activity data’ or ‘actigraphy’ or ‘accelerometer’) and(‘machine learning’ or ‘deep learning’ or ‘artificial intelligence). The step-by-step search method and filtering of the article stage-wise are depicted in detail in Figure 1.

In the first step of the search, 389 items were discovered, whereas 202 of these articles were removed due to duplicate entries. A further modification was done to match the titles and abstracts of the retrieved publications. As a result, the set was reduced to 98 items at this stage by discarding 89 articles that did not fulfill the targeted objectives. However, this collection also included publications that employed a variety of approaches to diagnose ADHD, including PPG, Multimodal, MRI, MEG, and others. However, the use of activity data for identification is a topic of concern. As a result, the other methods were abandoned, and ten publications based on activity data were scanned. Indeed, upon further analysis, five were deemed to be suitable and were chosen as the final group for literature analysis. Table 1 shows a detailed comparison of these articles based on different parameters.

Indeed, the literature review has certain inadequacies that lead to the current investigation through this paper. Some of the few drawbacks that are of concern are:Only one study out of five has utilized an adult dataset to diagnose ADHD.The majority of the studies relied on private datasets, which necessitated a significant amount of effort and time for data gathering and processing.On the activity datasets, only five studies have used machine learning approaches.The machine learning models applied to the activity dataset even did not provide many reliable and precise results.

As a result of these limitations, this study aims to improve the precision with which activity data can be utilized to distinguish ADHD patients from other clinical controls. In addition, there are relatively few publicly available datasets in this domain, and via this, a new publicly available dataset has been examined.

## 3. Methods and Materials

This section provides an overview of the proposed framework for the identification of ADHD among clinical controls. Figure 2 illustrates the proposed framework, and based on the dataset, this paradigm is useful for classifying patients with ADHD or Non-ADHD. In the first stage, data is gathered from varied sources, which consist of details of patients containing information regarding their background and medical history. Moreover, it takes as input the results of computerized tests and the patient’s activity log using wearable devices. Then, the relevant data pertinent to the goals are chosen. Following the selection of necessary data, it is processed through statistical feature extraction and selection, using principal component analysis. Afterwards, the analysis is performed using several classification algorithms to precisely choose an optimal algorithm based on various performance metrics for accurate identification of ADHD patients.

### 3.1. Data Acquisition

The present work makes use of a publicly available ‘Hyperaktiv’ dataset [1] for accurate classification of ADHD and non-ADHD patients. It comprises health, activity, and heart rate data from adult ADHD patients and other clinical controls. It is accessible at https://osf.io/3agwr (accessed on 2 April 2022) and is licensed under Creative Commons Attribution-Noncommercial 4.0 International (CC BY-NC 4.0) [1]. For acquisition of this dataset,103 patients are enrolled out of which 51 patients have been diagnosed with ADHD and 52 with other disorders termed as clinical controls. The information gathered includes recordings of motor activity, heart rate, results of a computerized test as well as multiple diagnostic and clinical assessments. The activity log of one of the patients diagnosed with ADHD is depicted in Figure 3, which has been analyzed over a span of 24 h. The movement activity is measured in the Inertial Measurement Unit (IMU) using a wrist-worn actigraph device (Actiwatch, Cambridge Neurotechnology Ltd., England, model AW4). It consists of a piezoelectric accelerometer configured to record the integration of movement strength, amount, and duration in the x, y, and z-axes movements greater than 0.05 g at a sampling frequency of 32 Hz [1,30]. It is clear from the below figure that patients diagnosed with ADHD are more hyperactive during early mornings and in the evenings. The activity variation has a pivotal role in the diagnosis of ADHD because of the time-dependent nature of ADHD hyperactivities [31]. 

In this dataset, all the patients are diagnosed by two experienced and qualified psychiatrists using the Mini-International Neuropsychiatric Interview (MINI Plus, version 5.0.0) [32]. The clinical assessments are done by an automated and digital neuropsychological response test commonly known as CPT-II (Conner’s Continuous Performance Test) [33], which is used to access problems with attention. Apart from activity data, the present dataset also contains ECG-based heart rate data, which have been recorded with a compact battery-powered chest-worn device that allows for mobility and extensive recording times. This type of heart rate data can be used to determine the Heart Rate Variability (HRV) [34], which is a representation of the time difference between successive heartbeats. In the present data, a total of 80 patients provided heart rate recordings, with 38 ADHD sufferers logging their heart rates for an average of 20.5 ± 3.9 h, and 42 clinical controls logging their heart rates for an average of 21 ± 4 h [1].

Furthermore, the feature’s file consists of pre-extracted attributes that are then used for training and testing of given machine learning models being employed in this work. The file consisting of patient information contains 32 different attributes for each of 103 patients. For visual representation and analysis, the software utilized is open source cloud computing environment Google Colaboratory utilizing Python 3.7.13 [35]. Figure 4a represents the count of the number of males and females in the study. Figure 4b represents the different age patterns of the patients involved in the study, where 1 = 17–29 yrs, 2 = 30–39 yrs, 3 = 40–49 yrs, and 4 = 50–67 yrs. Figure 4c,d represent the number of patients diagnosed with ADHD and other disorders, including attention deficit disorder (ADD). Figure 4e picturizes the output for the CPT-II test results for the Montgomery and Asberg Depression Rating Scale (MADRS) [36], which consists of 21 items that access the intensity of depression. Figure 4f represents the Wender Utah rating scale (WURS) [37], which evaluates the prevalence and severity of childhood ADHD symptoms.

### 3.2. Feature Extraction

In the present work, 788 statistical features have been employed which are extracted from the activity data of the patients. These features are listed in the features file of the Hyperaktiv dataset [1]. The features are extracted using the open-source python package tsfresh [38], which aids in the feature engineering of the time-series data of the activity dataset [39]. These features are associated with standard deviation, variance, skewness, kurtosis, root mean square value, entropy and other related metrics.

Since the number of the features present in the dataset file are cumbersome, so to get a brief overview some of these features are shown graphically using different types of plots. Figure 5a presents a violin plot of the absolute energy of the activity data. The standard deviation ranges are shown using a histogram in Figure 5b. While Figure 5c,d analyze the kernel density estimator for kurtosis and skewness. The values of autocorrelation and ricker’s continuous wavelet transform are represented in Figure 5e,f respectively. Lastly, Figure 5g,h represent the fast Fourier transform coefficient values and permutation entropy values using kernel density estimator and histogram plots, respectively.

### 3.3. Feature Processing

This step entails the processing of the relevant features related to the activity recordings. Due to the small dataset size, training and testing would not accurately reflect a machine learning model’s generalization capabilities and would unnecessarily diminish its predictive power. Therefore, Principal Component Analysis (PCA) has been employed in this work in order to find significant features having strong patterns for accurate identification of patients with ADHD. PCA is an unsupervised dimensionality reduction approach. In this approach, by lowering the variance, the strong patterns in the provided dataset can be extracted. It is a statistical technique that turns observations of correlated features into a set of linearly uncorrelated data through orthogonal transformation. In this way, the newly altered features that are discovered are called the principal components. The reduced principal component features are either less than or equal to the features initially passed as an input. The purpose of PCA is to keep as much information as feasible while reducing the number of variables in a data set [40]. The various steps involved in PCA are standardizing the range of variables, calculating the covariance matrix, and finally determining the eigen vectors and eigen values from the matrix [41]. Dimensionality reduction is accomplished in this dataset by selecting eigenvectors to account for a specific proportion of variance in the original data. The default variance is 0.95. The data is centered rather than being standardized. PCA is calculated using the covariance matrix instead of the correlation matrix. A maximum of five attributes are selected in the formation of modified attributes. Moreover, all the attributes are considered to be equally important and are retained in the converted ranked attributes.

To investigate the accuracy further, the dataset size is resampled. It creates a random subset of the dataset, with or without replacement. The entire sample needs to be fitted into the memory perfectly. It allows the calculation of standard errors and confidence intervals easily. The confidence intervals become more reliable as the sample size grows. However, it increases the risk of overfitting noise in the data. The consequences of this problem can be reduced by combining the resampling method with the cross-validation procedure, which has been employed in the current work.

## 4. Model Selection

### 4.1. Machine Learning Techniques and Hyperparameter Tuning

To categorize patients with ADHD and other clinical controls, six different machine learning algorithms are employed which include Naive Bayes (NB), k-Nearest Neighbor (kNN), Support Vector Machine (SVM), C 4.5 Decision Tree, Random Forest (RF), and LogitBoost. The eventual aim is to create algorithms with the highest classification accuracy while using the fewest possible features possible. To boost the performance of these models, essential hyperparameters must be identified and tuned to exactly fit these machine learning models using given data for creating reliable and accurate models. Moreover, these parameters assist in finding the right balance of bias and variance and hence preventing the model from overfitting and underfitting. The machine learning models and hyperparameters tuning are implemented using WEKA 3.8.6. The hyperparameter tuning for varied chosen algorithms is as follows:

#### 4.1.1. Support Vector Machine (SVM)

SVM is a kernel-based machine learning model for classification and regression challenges. Due to its strong theoretical foundation and capacity to generalize, SVMs have become one of the most extensively used classification methods in recent years. SVM’s main goal is to separate classes in a training set by using a surface that maximizes the margin between them. It maximizes the generalization capabilities of the model [42]. This algorithm is typically used for linearly separable data, but it can also be used for non-linearly separable data using the kernel approach.

In this work, the type of SVM used is C-SVC, where C stands for the penalty parameter of the error term, and the cost value is taken as 1. It strikes a balance between the need for a smooth decision border and the accuracy with which the training points are identified. The batch size is indeed 100, and the cache is 40 MB. The termination criteria tolerance (eps) is 0.001. The type of kernel employed in this model is the radial basis function, and the degree of the kernel is set to three. The seed value is taken as 1 and the shrinking heuristics are configured to true.

#### 4.1.2. Naive Bayes (NB)

The Naive Bayes classifier is a probabilistic classifier based on Bayes’ theorem, which states that each feature contributes equally to the target class. The NB classifier assumes that each feature is independent of the others and does not interact with them, implying that each feature contributes equally to the probability of a sample belonging to a specific class [43]. It is simple to use and compute and performs well on large datasets with many dimensions. This algorithm works best when associated features are removed from the model since they are voted twice in the model. It is noise-robust and ideal for real-time applications [43].

For this classifier, the preferred number of instances that are processed in one batch is 100. The ‘use kernel estimator’ is set to false, so normal distribution is used for numeric attributes. The ‘use supervised discretization’ value and ‘do not check capabilities’ are both initialized to false.

#### 4.1.3. C 4.5 Decision Tree 

C4.5 is based on the information gain ratio being evaluated by entropy. The test features are picked using the information gain ratio measure at each node in the tree. These measures are known as feature or attribute selection measures. The test feature for the current node is the attribute with the highest information gain ratio. The proportion of observations to total observations is known as the Information Gain Ratio (IGR) [44]. When a data set is large, the decision tree is pruned, which implies that extraneous branches that are insignificant in the overall computation are deleted.

In this classifier, the batch size and seed parameters are both set to 100 and 1, respectively. When the ‘collapse tree’ property is set to True, the sections of the tree that do not minimize training error are deleted. The value of the number of folds is taken as three, which defines the amount of data used for reduced error pruning where two folds are used for tree growth and one fold is utilized for pruning. While the confidence factor is set to 0.25, the ‘subtree raising’ is considered true when pruning. The minimum number of the object variable is set to two. The MDL correction is utilized when discovering splits on numeric attributes.

#### 4.1.4. Random Forest (RF)

Random Forest is based on ensemble learning, which is a way to integrate multiple classifiers to solve complicated problems and enhance the effectiveness of the algorithm. It is a classifier that combines several decision trees on different subsets of a dataset and averages their results to increase the dataset’s predictive accuracy. The random forest collects forecasts from each tree and predicts the ultimate output based on the majority votes of projections, rather than depending on a single decision tree. The more the number of trees in the forest, the higher the accuracy and lower the risk of overfitting. It predicts output with good accuracy and runs quickly even with a huge dataset even if data values are missing [45].

In RF classifier, the bag size percentage, number of iterations, and batch size are all set to 100 by default. The maximum depth and number of iterations of the tree are both set to 0. To create the ensemble, there is a need for one execution slot. The “Store out of Bag Predictions” is tuned to false and the seed value is fixed to one.

#### 4.1.5. k-Nearest Neighbor (kNN)

kNN is a case-based classification algorithm that keeps all of the training data. It is a simple but efficient classification approach that has been proved to be one of the most powerful techniques in text categorization. A model is created for kNN to improve its efficiency while keeping its classification accuracy. To classify a data record t, the k closest neighbors are obtained, forming a t neighborhood [46]. However, in order to apply kNN, the proper value for k is chosen, as the performance of classification is strongly dependent on this number. There are a few ways to figure out the k value, but one simple option is to run the algorithm numerous times with different inputs.

In kNN, the number of neighbors (k) is chosen to one. The batch size is one hundred. In the nearest neighbor search algorithm, the linear NN Search algorithm is employed in the analysis. When conducting cross-validation, the mean absolute error is utilized. The window size parameter is set to 0, implying that there is no limit on the number of training instances. The distance weighting and cross-validate are set to false.

#### 4.1.6. Logit Boost (LBoost)

LogitBoost is a popular boosting type for categorizing binary and multi-class data. It is an additive tree regression method that reduces the logistic loss. The boosting algorithm for LogitBoost is deduced from the statistical approach in which the loss, the function model and the optimization strategy are all three essential components. This algorithm uses a binomial log-likelihood to modify the loss function linearly. In this, outliers and noise are less noticeable [47].

For this classifier, the Zmax response threshold is set to 3.0. All of the seed, shrinkage, pool size and number of threads variables are set to one. The chosen base classifier is the Decision Stump. The number of iterations is set to ten. The likelihood threshold is set equal to −1.79769313, while the weight threshold and batch size are adjusted at 100.

### 4.2. Performance Evaluation

The performance of an algorithm for a specific task is evaluated using different metrics. For this purpose, the confusion matrix is widely used, which is a four-quadrant table with rows and columns for presenting the classifier’s classification results. It aids in the compilation of performance evaluation measures. The four quadrants in the matrix consist of True Positive (TP), True Negative (TN), False Positive (FP) and False Negative (FN). Various performance metrics derived from the confusion matrix, which are employed in the present work, include accuracy [48], sensitivity [48], specificity [48], F-measure [49] and AUC [50]. These metrics are illustrated as per the following details:(1)$$\mathrm{Accuracy}=\frac{TP+TN}{TP+TN+FP+FN}$$
(2)$$\mathrm{F-measure}=\frac{\;2*\mathrm{Precision}*\mathrm{Recall}}{\mathrm{Precision}\;+\;\mathrm{Recall}}$$
(3)$$\mathrm{Sensitivity}=\frac{TP}{TP+FN}$$
(4)$$\mathrm{Specificity}=\frac{TN}{TN+FP}$$

AUC is a cumulative measure of performance that takes into account all possible classification thresholds. AUC is a two-dimensional region beneath the complete AUC curve from (0, 0) to (1, 1). The AUC value varies from 0 to 1. The AUC of a model whose predictions are 100 percent incorrect is 0.0, whereas the AUC of a model whose predictions are 100 percent correct is 1.0 [50].

## 5. Results 

This section demonstrates the results of multiple classification algorithms to ensure the relevancy of the proposed methodological approach for the detection of ADHD. The different classification algorithms chosen for this work are C4.5, kNN, LBoost, NB, SVM, and RF. For all the algorithms, ten-fold cross-validation is used to analyze the result. The performance of the employed classification algorithms is presented in Table 2 in terms of accuracy, sensitivity, specificity, F-measure and AUC.

Furthermore, the comparison is drawn graphically for accuracy values among given ML algorithms in Figure 6, which also shows the superiority of SVM among other algorithms for accurate identification of ADHD patients, whereas Figure 7, Figure 8 and Figure 9 display the performance of given ML algorithms in terms of sensitivity, specificity and F-measure values of ADHD and non-ADHD classes. Figure 10 presents AUC curves for SVM, RF, C4.5, kNN, LB and NB classifiers. 

## 6. Discussion

### 6.1. This Work

It is evident from Table 2 that the SVM classifier outperforms with an accuracy of 98.43% and the Naïve Bayes algorithm exhibits the lowest accuracy of 80.39% among other classification algorithms. The other performance metrics values such as F-Measure, sensitivity, specificity, and AUC for SVM are 98.42%, 98.33%, 98.56%, and 0.983, respectively. However, the evaluation results of RF and kNN are quite close to the performance of SVM. They both have showcased accuracies of nearly 97.25% and 97.65%, respectively. While the AUC for RF is the maximum among all the algorithms, which is 0.999. According to the findings of the evaluation, the NB and LBoost do not appear to be appropriate classifiers for identifying ADHD as they exhibit much lower values of performance metrics in comparison to the other models.

Figure 7, Figure 8 and Figure 9 pictorially represent how well these metrics performed on various models to differentiate between the two classes of classifying the ADHD patients from clinical controls. As expected, SVM classifier has performed well for this task in terms of sensitivity, specificity and F-measure having class ADHD values of 100%, 100% and 98.54%, respectively.

As per Figure 10 and Table 2, SVM exhibits a significantly high value of AUC along with Random Forest having values of 0.983 and 0.999, respectively. On the other hand, NB classifier shows a poor AUC of 0.889 only. Therefore, the thorough analysis of given ML algorithms reveals that SVM algorithm performs better amongst other algorithms with maximum values of accuracy, sensitivity, specificity, F-measure and an optimal value of AUC, which makes it a suitable choice for accurate identification of ADHD patients among other clinical controls.

### 6.2. Contributions and Limitations

This research work utilizing activity data provides an efficient machine learning model, which can efficiently differentiate between ADHD and clinical controls with an accuracy of 98.43% and AUC of 0.983. However, this study also poses some limitations. The key drawback is the scarcity of data samples due to clinical data collection difficulties. Another issue is that the subtype of ADHD was not taken into account due to the diversity of the ADHD population [11]. Additionally, new strategies for detecting ADHD, such as transfer learning, can be applied [51]. Deep learning algorithms can also be used to distinguish ADHD patients from clinical controls using EEG data [52]. Moreover, indicators such as pupil size can be used to determine whether or not a patient has ADHD [16]. In connection to ADHD, the gender and age difference factors could also be used [12]. Furthermore, the relationship between heart rates and ADHD could also be investigated, and the validity of computerized tests used to diagnose ADHD could be tested.

### 6.3. Future Scope

Future research plans include looking at ADHD in context with the patients’ heart rates. The research should be expanded to look into the causes and effects of additional conditions such as anxiety, bipolar disorder, unipolar disorder, and depression on ADHD. Additionally, the validity of the computerized tests that are currently used to assess the prevalence of ADHD must be examined.

## 7. Conclusions

In this study, the optimal autonomic model for the diagnosis of ADHD is determined using six machine learning algorithms. The phenomenal performance of classifiers in this work is attributed to the computed feature selection approach and most importantly, the identification and combination of complementary features using PCA. The SVM model proposed is the most accurate classifier, classifying at 98.43% accuracy with high sensitivity, specificity and F-measure. The overall analysis of the proposed work clearly reveals that PCA based SVM approach is a highly suited method for accurate identification of ADHD patients. The overall findings provide new information regarding accurate machine learning algorithms, which can reduce misinterpretation and can be used to assess treatment effectiveness.

## Figures and Tables

**Figure 1 brainsci-12-00831-f001:**
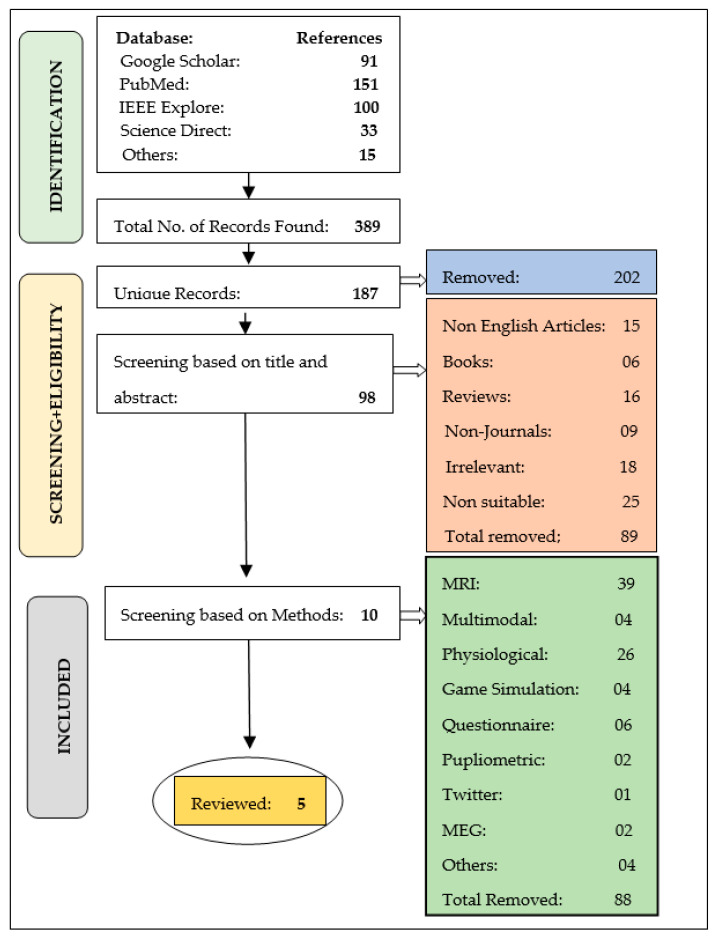
Flow diagram for filtering of articles.

**Figure 2 brainsci-12-00831-f002:**
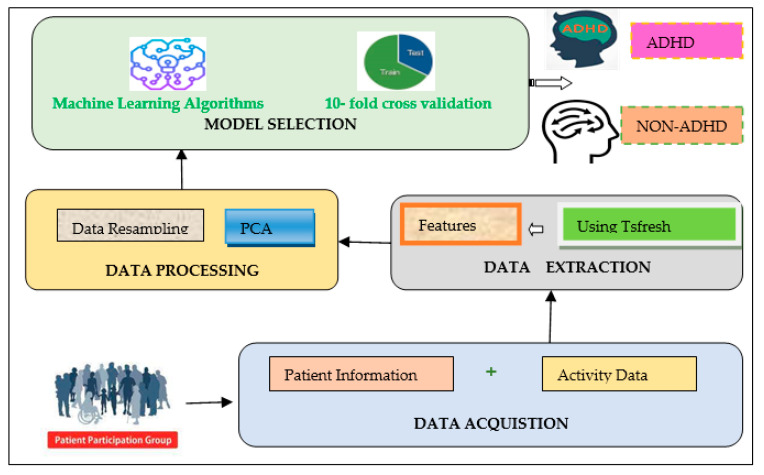
The proposed framework.

**Figure 3 brainsci-12-00831-f003:**
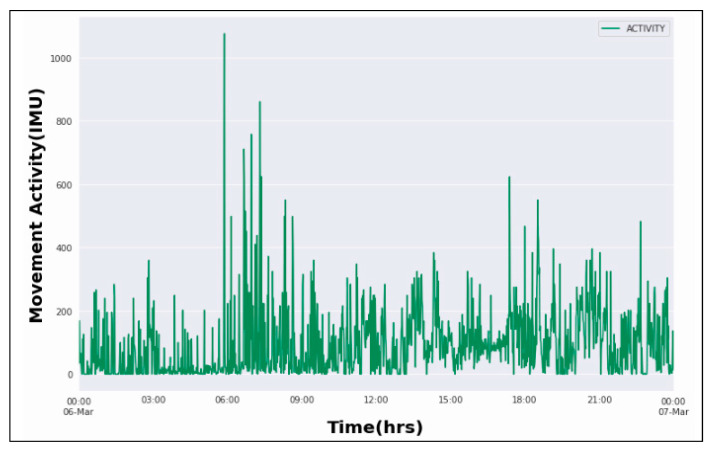
Activity variation in ADHD patient with respect to time.

**Figure 4 brainsci-12-00831-f004:**
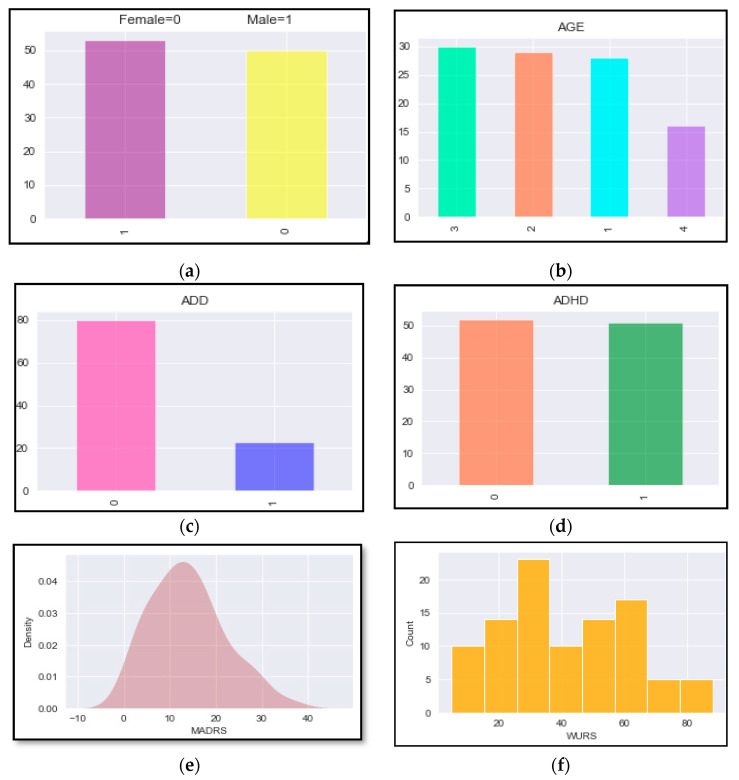
Patient information in terms of different parameters: (**a**) males vs. females; (**b**) age group of patients; (**c**) ADD vs. non-ADD; (**d**) ADHD vs. non-ADHD; (**e**) MADRS values; and (**f**) WURS values.

**Figure 5 brainsci-12-00831-f005:**
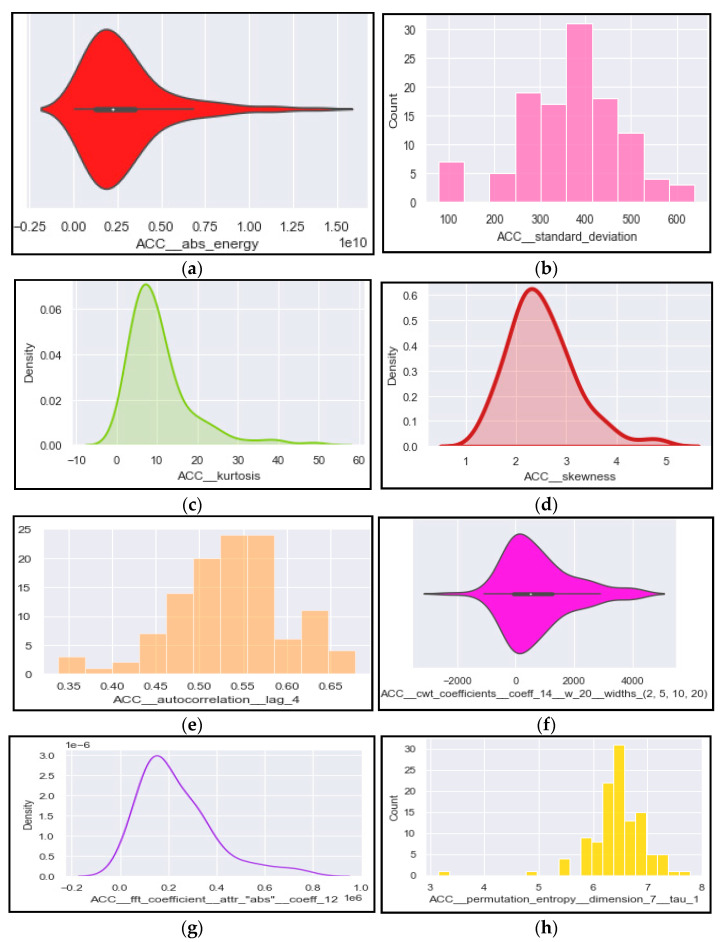
Visual representation of various activity related features: (**a**) absolute energy; (**b**) standard deviation; (**c**) kurtosis; (**d**) skewness; (**e**) autocorrelation values; (**f**) continuous wavelet transform (**g**) fast Fourier transform; and (**h**) permutation entropy.

**Figure 6 brainsci-12-00831-f006:**
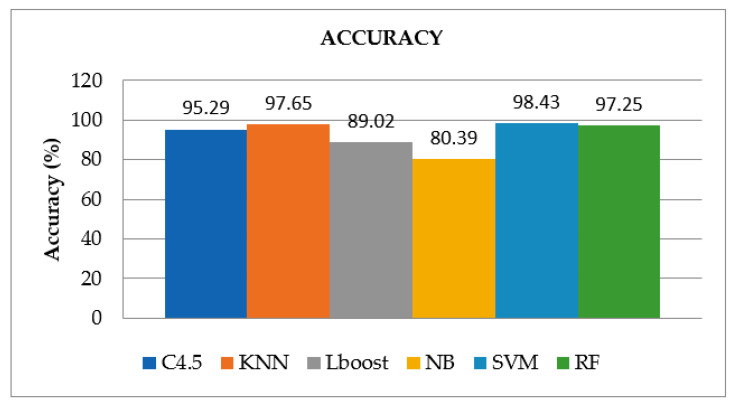
Performance comparison of ML algorithms in terms of accuracy.

**Figure 7 brainsci-12-00831-f007:**
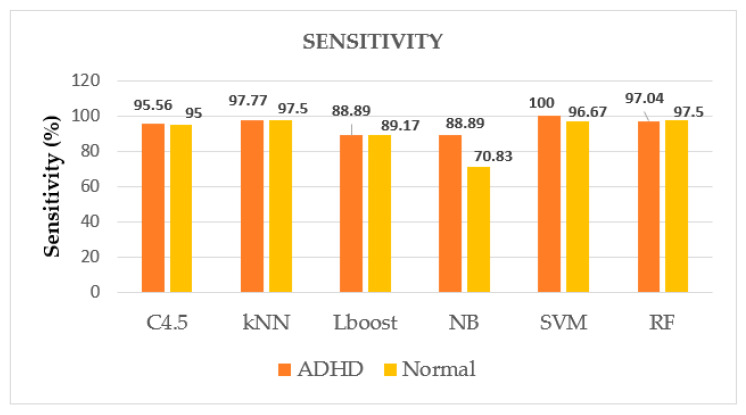
Performance of ML algorithms in terms of sensitivity of ADHD and non-ADHD classes.

**Figure 8 brainsci-12-00831-f008:**
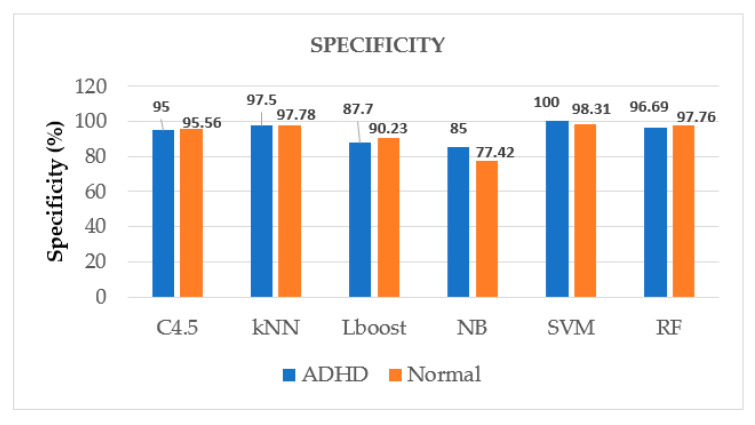
Performance of ML algorithms in terms of sensitivity of ADHD and non-ADHD classes.

**Figure 9 brainsci-12-00831-f009:**
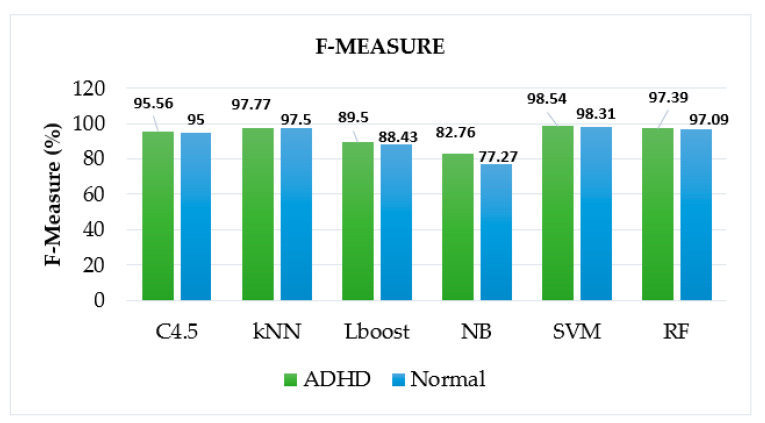
Performance of ML algorithms in terms of sensitivity of ADHD and non-ADHD classes.

**Figure 10 brainsci-12-00831-f010:**
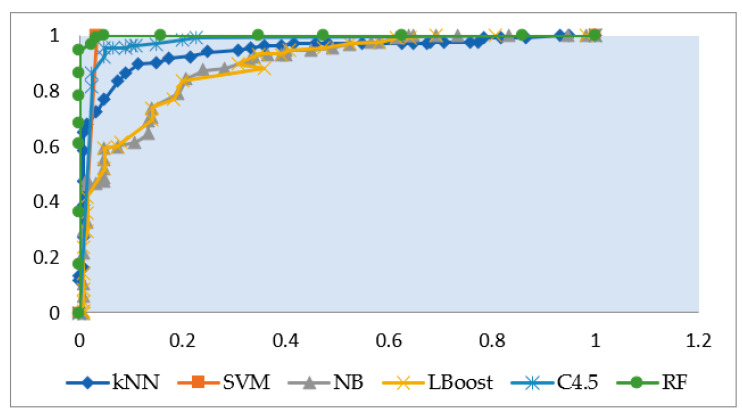
Area under the ROC curve (AUC) for different models.

**Table 1 brainsci-12-00831-t001:** Description of studies that used actigraphy and accelerometer data.

S.No	Reference	Year	Dataset	Age Group	Public/Private	Method	Features	Model	Validation Approach	Highest Accuracy
1	Munoz-Organero et al. [27]	2018	22 school children with ADHD = 11, Paired Controls = 11	6–15 years	Private	Two trial axial accelerometers: one on the wrist of the dominant arm and the other on the axle of the dominant leg	2D acceleration images	Deep learning	4-fold cross-validation	CNN 87.5% with wrist sensor and 93.8% with axle sensor
2	Faedda et al. [28]	2016	155 children with ADHD = 44 ADHD + Depression = 21 Bipolar = 48 Controls = 42	5–18 years	Private	Belt worn actigraphs	28 metrics	Machine Learning	4-Fold cross validation	SVM 83.1%
3	Amado-Caballeroat et al. [29]	2020	148 children with ADHD = 73 Normal = 75	6–15 years	Private	Wrist Worn ActiGraph GT3x	End-to-End	Deep Learning	10 fold cross validation	CNN 98.6%
4	O’Mahony et al. [24]	2014	43 children with ADHD = 24 Normal = 19	6–11 years	Private	Two IMU one at the waist and the other at the ankle of the dominant leg	Inertial measurement Units	Machine Learning	Leave one out cross-validation	SVM 95.1%
5	Hicks et al. [1]	2021	103 patients with ADHD = 51 Normal = 52	17–67 years	Public	Wrist-worn Actigraph device	Feature extraction with tsfresh	Machine Learning	10 fold cross-validation	Random Forest gives 72%

**Table 2 brainsci-12-00831-t002:** Performance evaluation results of different classification algorithms.

S.No	Model	Accuracy (%)	Sensitivity (%)	Specificity (%)	F-Measure (%)	AUC
1	C4.5	95.29	95.28	95.28	95.28	0.973
2	kNN	97.65	97.64	97.64	97.64	0.975
3	LBoost	89.02	89.03	88.96	88.99	0.941
4	NB	80.39	79.86	81.21	80.02	0.889
5	SVM	98.43	98.33	98.56	98.42	0.983
6	RF	97.25	97.27	97.23	97.25	0.999

## Data Availability

The data is available at https://osf.io/3agwr (accessed on 2 April 2022).

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
