# Peer review of "Accurate Identification of ADHD among Adults Using Real-Time Activity Data"

_brainsci, 2022, doi:10.3390/brainsci12070831_

Round 1
Reviewer 1 Report
Undoubtedly, attention deficit hyperactivity disorder is a neurodevelopmental disorder that affects millions of children and typically persists into adulthood. Authors present in this paper, PCA based SVM approach appears to be an effective choice for accurate identification of ADHD patients among other clinical controls using real-time analysis of activity data.
My comments to the article are as follows:
- I propose to extend the introduction in the article by referring to the fact that in practice research is also carried out to analyze the EEG signal in the field of concentration of attention in mothers and children. In this regard, the following article can be cited: EEG Analysis and Neurofeedback Therapy of Concentration Problems in Mother and Child, published in Advances in Soft Computing, Springer, 2021.
- The diagram in fig. 1 should be modified as it is difficult to read. Diagram no. 2 is similar. Pictures require professional graphic correction.
- Chapter 3 should not start with a Figure, but with the content. It needs to be expanded.
- Chapter 4 needs to be expanded to describe each method in more detail. The current description is general.
- There is no axle description for Fig. 3. Similarly, the following graphs in the article are not described.
- The Results section should be separate from Discussion. There should be a specific discussion of the results.
- Charts / all graphic forms should be made with greater care. Currently they are not well developed. There is no standardization.
Reviewer 2 Report
Dear Authors; I found this work interesting exploration of different machine learning based method to diagnose ADHD status using activity data. IT needs some "extra work" to make it publishable. Regards, P.S.
[1] Writing:
1-1 Add State, Country to the author affiliation !
1-2 Format References to MDPI . For example, for published articles, years are in bold.
1-3 Add the list of used Abbreviations in the work right before Reference section for readers easy access. Example: Abbreviations: ADHD: Attention Deficiency Hyperactivity Disorder, ....
1-4 Break down the Results and Discussion section into two section. The current situation is strange and contrary to conventional standard IMRD format. So, "5.Results".
1-5 Rewrite the Discussion and Conclusion sections again. So, 6. Discussion, 6.1. This Work, 6.2. Contributions and Limitations, 6.3. Future Work, 6.4. Conclusion
[2] Statistical:
2-1 In 4 out of 5 sample datasets in the work you have participants aged <18 ( not adults). This is contrary to the claim in the Abstract on being "adults" . Explain. Modify the text wherever it is necessary.
2-2 Which statistical software did you use to (i) conduct the analysis, (ii) plot the Figures in the paper ? Write them and cite them in the references.
2-3 Equation (5) in page 10 is wrong. AUC is not interval. Write down its statistical formulae there.
2-4 Which potential confounders the study data analysis did not consider them ? Check study 5 data sets. Write them in the limitations section.
Round 2
Reviewer 1 Report
Dear Authors,
Thank you for the changes made to the article.
As it stands, I have no more comments, although I would further work on some standardization of data presentation in charts.
Reviewer 2 Report
Dear Authors, most of my main concerns were addressed satisfactorily. Regards.